# An Outbreak of Severe Neonatal Pneumonia Caused by Human Respiratory Syncytial Virus BA9 in a Postpartum Care Centre in Shenyang, China

Bing Wang,[a,b] Jingjing Song,[a] Jinhua Song,[a] Naiying Mao,[a] Jiayuan Liang,[c] Ye Chen,[b] Ying Qi,[b] Lina Bai,[b] [ID]Zhibo Xie,[a] Yan Zhang[a]

[a]National Health Commission (NHC) Key Laboratory of Medical Virology and Viral Diseases, National Institute for Viral Disease Control and Prevention, Chinese Center for Disease Control and Prevention, WHO WPRO Regional Reference Measles/Rubella Laboratory, Beijing, China
[b]Shenyang Prefecture Center for Disease Control and Prevention, Shenyang, China
[c]Liaoning Provincial Center for Disease Control and Prevention, Liaoning, China

Bing Wang and Jingjing Song contributed equally to this article. The order was determined by authors after negotiation.

**ABSTRACT** Human respiratory syncytial virus (HRSV) is a major pathogen of lower respiratory tract infections in children (<5 years) and older individuals, with outbreaks mainly reported among infants in hospital pediatric departments and intensive care units (ICUs). An outbreak of severe neonatal pneumonia occurred in a postpartum center in Shenyang city, China, from January to February 2021. In total, 34 respiratory samples were collected from 21 neonates and 13 nursing staff. The samples were screened for 27 pathogens using a TaqMan low-density array, and 20 samples tested positive for HRSV, including 16 neonates and 4 nursing staff samples. Among the 16 hospitalized neonates, seven were admitted to an ICU and nine to general wards. Four of the nursing staff had asymptomatic infections. To investigate the genetic characteristics of the HRSV responsible for this outbreak, the second hypervariable region (HVR2) sequences of the G gene were obtained from six neonates and two nursing staff. Phylogenetic analyses revealed that all eight sequences (SY strains) were identical, belonging to the HRSV BA9 genotype. Our findings highlight the necessity for strict hygiene and disease control measures so as to prevent cross-infection and further avoid potential outbreaks of severe infectious respiratory diseases.

**IMPORTANCE** Human respiratory syncytial virus (HRSV) is one of the leading causes of acute lower respiratory infections (ALRI) worldwide. In this study, we first reported an outbreak of severe neonatal pneumonia caused by HRSVB BA9 at a postpartum care center in mainland China. Among 20 confirmed cases, 16 were hospitalized neonates with 7 in the ICU ward, and the other four were nursing staff with asymptomatic infections. Our findings highlighted the importance of preventing cross-infection in such postpartum centers.

**KEYWORDS** BA9 genotype, human respiratory syncytial virus (HRSV), neonatal pneumonia, outbreak, postpartum care center

**H**uman respiratory syncytial virus (HRSV) is one of the most common pathogens of acute lower respiratory infections (ALRI) in young children under 5 years of age, usually requiring hospitalization and mainly causing pneumonia and bronchiolitis and even death in severe cases (1, 2). The main manifestations of ALRI caused by HRSV are a stuffy nose, runny nose and fever, cough, shortness of breath, and respiratory failure (2). Its serious disease burden has aroused great public concern. In 2015, an estimated 33.1 million children infected with HRSV-related ALRI were reported worldwide, of which 3.2 million children were hospitalized (3).

HRSV is a negative-sense, single-stranded RNA virus of the genus *Orthopneumovirus* and the family *Pneumoviridae* (4). According to antigenic and genetic differences, HRSV is divided into the HRSVA and HRSVB subtypes (5). Based on the second hypervariable region

**Ad Hoc Peer Reviewer** [ID]Leyda Abrego, Instituto Conmemorativo Gorgas de Estudios de la Salud; Hirokazu Kimura, Gunma Paz University

Address correspondence to Yan Zhang, zhangyan@ivdc.chinacdc.cn.

The authors declare no conflict of interest.

**TABLE 1** Characteristics of nine severe cases of human respiratory syncytial virus infection in patients hospitalized in Shenyang Children's Hospital

| Case no. | Age (days) | Gender (M/F)[a] | Date of disease onset | Pneumonia on CT[b] scan | Diagnosis | Nasal | Length of cough | Presence/ length of fever | WBC[c] (× 10⁹/L) | Respiratory failure | Heart failure | Hospitalized | ICU |
|---|---|---|---|---|---|---|---|---|---|---|---|---|---|
| 1 | 27 | M | Jan 17 | Yes | ALRI | No | 4 days | No | 11.4 | Yes | No | Yes | Yes |
| 2 | 26 | F | Jan 23 | Yes | ALRI | No | 10 days | No | 9.7 | Yes | No | Yes | Yes |
| 3 | 26 | F | Jan 26 | Yes | ALRI | No | 7 days | No | 9.7 | Yes | Yes | Yes | Yes |
| 4 | 17 | M | Jan 28 | Yes | ALRI | 4d | 3 days | No | 7.7 | Yes | No | Yes | Yes |
| 5 | 28 | M | Jan 29 | Yes | ALRI | 2d | 2 h | No | 9.9 | Yes | No | Yes | Yes |
| 6 | 22 | M | Jan 30 | Yes | ALRI | 3d | 3 days | No | 9.2 | Yes | No | Yes | Yes |
| 7 | 12 | M | Feb 1 | Yes | ALRI | No | 2 days | No | 9.2 | Yes | No | Yes | Yes |
| 8 | 27 | M | Jan 30 | Yes | ALRI | 1d | 4 days | 1 day | 12.93 | No | No | Yes | No |
| 9 | 23 | M | Feb 3 | Yes | ALRI | 2d | 2 days | No | 6.3 | No | No | Yes | No |

[a]M, male; F, female.
[b]CT, computed tomography.
[c]WBC, white blood cells.

(HVR2) located in the C-terminal domain of the G gene, subtypes HRSVA and HRSVB can be further classified into 15 genotypes and 30 genotypes, respectively (6–8). In recent years, the BA9 genotype with a 60-bp repeat insertion in the HVR2 region of G protein has gradually become the predominant genotype globally (9).

Outbreaks of HRSV infection usually occur in general pediatric wards and neonatal intensive care units (ICU) but have occasionally been reported in postpartum care center settings (1, 10–13). In recent years in China, an increasing number of postpartum women seek professional care in postpartum care center in the first few months after discharge from maternity hospitals, so neonates may be exposed to some potential risk factors for respiratory disease infection in the postpartum centers.

This study describes an outbreak of neonatal pneumonia in a postpartum care center in Shenyang city, Liaoning province, China, during January and February 2021. Clinical samples were collected from neonates and nursing staff to identify the etiological agent and determine its origin. Screening of the samples indicated that the outbreak was caused by HRSV. HVR2 fragments of the G gene of SY strains of HRSV were obtained to clarify the genetic characteristics of the virus.

## RESULTS

**Epidemiological investigation.** From 17 January to 3 February 2021, an outbreak of neonatal pneumonia was reported in a postpartum care center in Shenyang city, China. In total, 16 out of 21 neonates from the Shenyang postpartum care center were hospitalized for clinical treatment because of symptoms of ALRI. Among the 16 hospitalized ALRI cases, seven neonates, including premature twins, were diagnosed with pneumonia and admitted to the ICU, and nine neonates had mild respiratory infection cases. The average age of these 16 neonates was 23 days after birth (range, 12 to 28 days). In addition, 4 of the 13 nursing staff members working in the postpartum care center during the outbreak belonged to lab-confirmed respiratory infection cases, and the average age of cases was 39 years (range, 24 to 47 years). No additional cases were reported after 3 February 2021.

The index case of this outbreak (case no. 1) occurred in the postpartum care center on 17 January 2021 with symptoms such as respiratory failure and was subsequently admitted to the hospital after 3 days onset. Then, case no. 2 in the same postpartum care center developed severe clinical symptoms 7 days after the onset of case no. 1, including a cough, shortness of breath, respiratory failure, and anemia. Case no. 2 and 3 were twins that were born prematurely, and case no. 3 presented similar symptoms to case no. 2 except for heart failure. Afterward, case no. 4, 5, 6, and 7 from the same center presented the same symptoms as the above cases, while all of the subsequent cases had relatively mild symptoms, mainly manifested as coughing and nasal congestion (Table 1). Seven of the neonates (case no. 1 to no. 7) were transferred to the ICU because of heart failure and respiratory failure, and all received intubation treatment. All neonatal cases recovered after treatment in hospital.

**Etiological identification.** To identify the cause of this outbreak, pharynx swabs or nasopharyngeal aspirates were collected from 21 neonates, while throat swabs were collected from 13 nursing staff. All of the samples were screened for 16 viruses, namely, adenovirus; human bocavirus; parainfluenza virus; respiratory syncytial virus; influenza virus; varicella zoster virus; Epstein-Barr virus; cytomegalovirus; human herpesvirus 6; human metapneumovirus; measles virus; coronavirus 229E, HKU1, NL63, and OC43; mumps virus; enterovirus; rhinovirus; and parechovirus, and 11 bacteria, namely, *Bordetella, Bordetella holmesii, Bordetella pertussis, Chlamydophila pneumoniae, Haemophilus influenzae, Klebsiella pneumoniae, Legionella pneumophila, Moraxella catarrhalis, Mycoplasma pneumoniae, Staphylococcus aureus,* and *Streptococcus pneumoniae.* Twenty of the 34 samples tested positive only for HRSV, and negative results were obtained for all other viruses and bacteria. The 20 HRSV-positive samples were from 16 hospitalized neonates and four nursing staff with asymptomatic infections. The other five asymptomatic neonates and nine nursing staff were negative for all 27 pathogens.

**Characterization of HRSV associated with this outbreak.** The HVR2 fragment of the G gene of HRSV (324 nucleotides) was successfully obtained from six neonates and two nursing staff. Homology analysis revealed that these eight sequences were 100% identical. Phylogenetic trees were constructed with the eight sequences from this study and HRSV B reference sequences downloaded from the GenBank database. These eight SY strains were clustered into the same branch as the BA9 genotype reference sequences. The sequences of the viruses identified in this study were most closely related (98% homology) to viruses detected in the Netherlands and Spain during 2018 and 2019 (Fig. 1a). The same results were obtained by performing a BLAST search of the SY sequences against the GenBank database.

Next, 83 representative BA9 HVR2 sequences originating from 26 countries between 2005 and 2019 were retrieved from the GenBank database. Phylogenetic analysis was performed on the SY strains and the 83 representative BA9 HVR2 sequences. The results showed that SY strains were clustered together with the BA9 viruses found in other countries between 2017 and 2019 (Fig. 1b; see also Table S1 in the supplemental material).

Compared with the BA9 reference strain NG-102-06 (GenBank accession number AB603467), there were five amino acid mutations (A271V, T276A, I281T, T290I, and T312I) in the HVR2 region of the G gene observed for all of the SY strains. The SY strains also possessed a mutation in a stop codon (from TAA to CAA), which led to a 7-amino-acid extension to the G protein (Q-R-L-Q-S-Y-A).

## DISCUSSION

From January to February 2021, an outbreak of severe neonatal pneumonia caused by the BA9 genotype of HRSV was reported in a postpartum care center in Shenyang city, China. The outbreak resulted in the hospitalization of 16 neonates, and of the nine severe neonatal cases, seven presented with respiratory failure, including a pair of premature twins who also presented with heart failure. This represents the first report of an HRSV-related severe neonatal pneumonia outbreak at a postpartum care center in mainland China.

The most common transmission routes of HRSV are via the respiratory tract and direct contact, and the virus is able to survive on surfaces such as countertops and cribs for hours (14). The HRSV incubation period ranges from 3 to 8 days (15). Although the index case was identified among neonates, the nursing staff or visiting relatives with asymptomatic HRSV infections were considered as the possible source of this outbreak, which could be supported by the following evidence. First, all of these neonates were transferred from different maternity hospitals after birth, and the onset of disease in the index infant was 21 days after moving to the postpartum care center, which is much longer than the HRSV infection incubation period. Therefore, it is likely that the index case was infected at the postpartum care center rather than the maternity hospital. Secondly, the investigation found that each neonate was cared for in a separate room but shared the same nursing staff. The nursing staff did not routinely wear masks and took care of different neonates without hand sanitation or changing gowns. HRSV is a very contagious infectious disease with an R0 of around

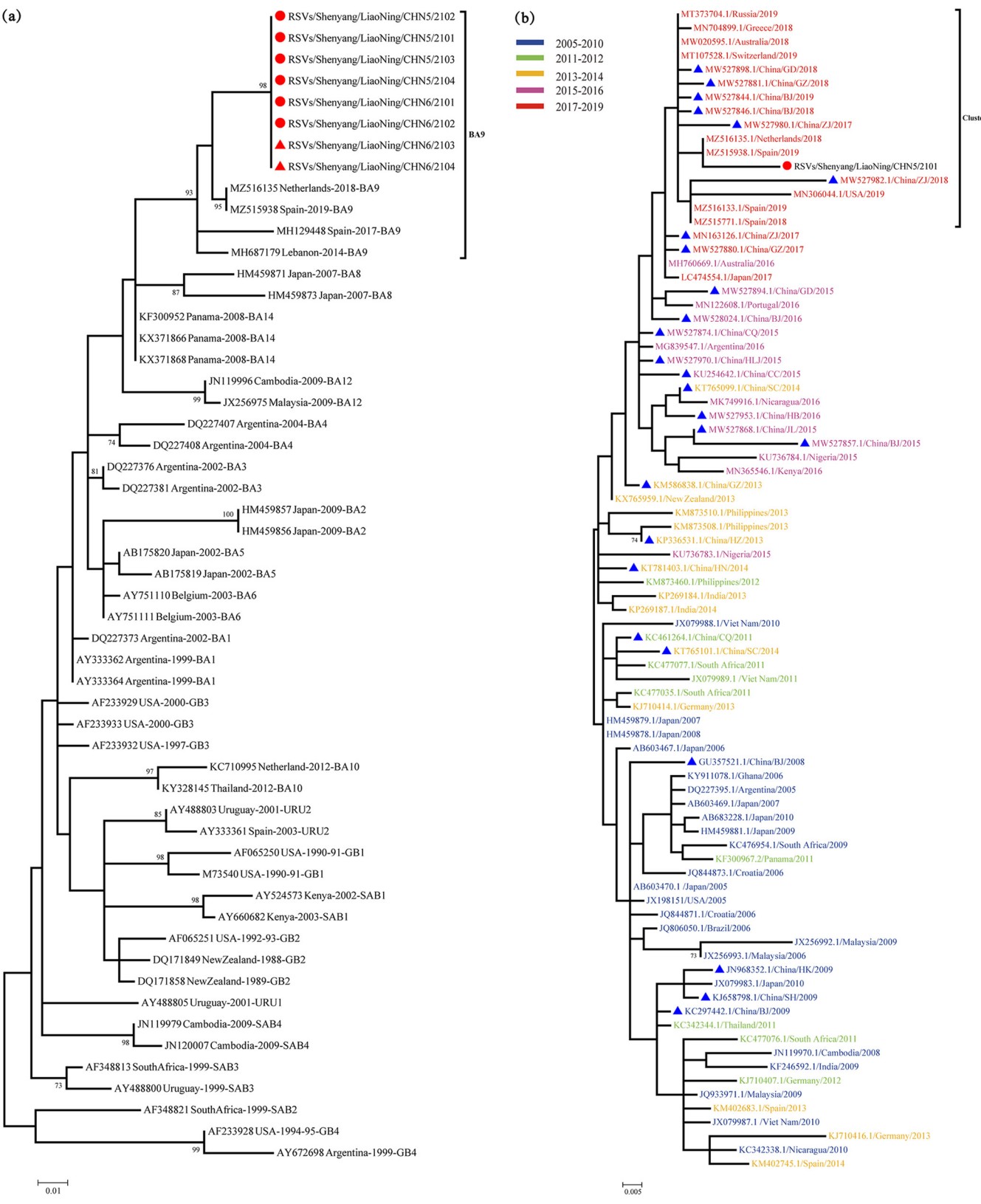

**FIG 1** Maximum likelihood phylogenetic tree of the entire coding region nucleotide sequence of HVR2 of the G gene of HRSV subgroup B isolated from the postpartum care center. (a) Red circles represent sequences from neonates; red triangles represent sequences from nursing staff. (b) Phylogenetic analysis of the sequences of SY strains from the outbreak in Shenyang (shown in red) and 83 genotypes of BA9 strains (2005 to 2019) retrieved from the GenBank database. Blue triangles indicate representative strains from China.

~1.0 to ~3.0 (16, 17). Based on this situation, the viruses might be carried by the nursing staff and transmitted to the neonate and then spread from one neonate to another. Thirdly, an investigation of the postpartum care center after the HRSV outbreak found that it is a relatively independent building, covering an area of 800 m² with 28 rooms across three floors. Though each room contains a bathroom and a baby cot, there were no disinfection records for any objects or rooms, the stored disinfectant had expired, and the rooms were poorly ventilated. In addition, routine monitoring measures for the health status of staff members were not performed. Accordingly, the postpartum center was considered to have a potential risk of cross-infection, and thus, it is inferred that this outbreak of neonatal pneumonia might be spread by nursing staff with asymptomatic infections. However, visiting relatives who might bring HRSV to the postpartum care center could not be ruled out. Unfortunately, samples were not available from visiting relatives, so there is no evidence to support this speculation.

Amino acid analysis of the SY strains identified in this study showed that there were five mutations in the HVR2 of the G protein gene, including A271V, T276A, I281T, T290I, and T312I. Previous studies have found the same amino acid mutation sites, and five simultaneous amino acid mutation sites were found in six identical sequences from patients hospitalized with community-acquired pneumonia (CAP) from China during 2018 (18). However, whether these five amino acid mutations found in the viruses associated with severe neonatal pneumonia could be beneficial for virus replication and transmission remains to be further studied. In addition, we identified a mutation in a stop codon of SY strains that resulted in a 7-amino-acid extension to the G protein. This was consistent with a previous report from the Observational United States Targeted Surveillance of Monoclonal Antibody Resistance and Testing of HRSV (OUTSMART-RSV) study, which analyzed viruses circulating in the United States during 2016 and 2017 (19). The function of these seven extended amino acids in the G protein remains to be determined, and this extended G protein has not previously been linked to the outbreaks of severe neonatal pneumonia. However, future research and surveillance to monitor the effect of this genetic modification on viral epidemiology, transmission, and disease severity are needed.

In conclusion, this outbreak of severe neonatal pneumonia that occurred in Shenyang of China was caused by HRSV genotype BA9. The postpartum center was considered to have a potential risk of cross-infection, and it is inferred that this outbreak of neonatal pneumonia may have been spread by nursing staff with asymptomatic infections. However, it could not be ruled out that visiting relatives may have brought HRSV to the postpartum care center. Our findings highlight the importance of strict hygiene and disease control measures, including wearing/changing masks, hand washing, and changing gowns between neonates, to prevent potential outbreaks of severe respiratory infectious diseases in such clinical settings. To the best of our knowledge, this is the first report to describe a severe neonatal pneumonia outbreak caused by the HRSV BA9 genotype in a postpartum care center in China.

## MATERIALS AND METHODS

**Ethics statement.** This study was approved by the second session of the Ethics Review Committee of the National Institute for Viral Disease Control and Prevention of the Center for Disease Control and Prevention (CDC) in China (IVDC 2018 no. 012). Written informed consent for the use of clinical specimens was obtained from all patients involved in this study or their guardians. This study did not involve human experimentation; the only human material used in this study was nasopharyngeal swab and throat swab specimens collected from suspected ALRI cases during an outbreak in Shenyang city of Liaoning province, China, from January to February 2021.

**Specimen collection.** During the outbreak period, nasopharyngeal swabs and throat swabs were collected from all neonates and nursing staff at the postpartum care center. All specimens were collected by epidemiology staff of Shenyang CDC and were transported in sterile containers with a cold package (controlled low temperature of 4℃) to the Institute for Viral Disease Control and Prevention for further analysis.

**RT-PCR and sequencing.** The viral nucleic acid was directly extracted from the clinical specimens using the Tianlong nucleic acid extraction kit (Tianlong Biotechnology, Xian, China) according to the manufacturer's instructions. The samples were screened for human respiratory pathogens, including 16 viruses and 11 bacteria using multiplex real-time reverse transcriptase PCR (RT-PCR) with the TaqMan low-density array (TLDA) kit (Thermo Fisher Scientific Inc., Waltham, USA). The subtypes of HRSV were further identified by in-house real-time RT-PCR (20). The second hypervariable region (HVR2) of the G gene of HRSV B subtype was amplified using a one-step reverse transcription-PCR kit (TaKaRa Biotechnology, Dalian, China) and the primer

pair GPB/F1 (21). The reaction conditions, as well as the purification and sequencing protocols, were as described previously (21, 22).

**Phylogenetic analysis.** Sequences were edited with Sequencher 5.0 (GeneCodes, Ann Arbor, MI, USA). Multiple sequence alignments and pairwise distance were determined using the MEGA program (version 5.0; Sudhir Kumar, Arizona State University). Phylogenetic trees were generated using MEGA with the maximum likelihood (ML) method. The reliability of phylogenetic inference was estimated using the bootstrap method with 1,000 replicates. Bootstrap values of ≥70% are shown.

**Data availability.** All sequences obtained in this study were submitted to the GenBank database under the accession numbers OM892937 to OM892944.

## SUPPLEMENTAL MATERIAL

Supplemental material is available online only.

**SUPPLEMENTAL FILE 1**, XLSX file, 0.01 MB.

## ACKNOWLEDGMENTS

This work was supported by the National Health Commission Major Public Health Project (ZDGW21-131031103000180005) and Key Technologies R&D Program of the National Ministry of Science (2018ZX10713002).

We declare that no competing interests exist.

Bing Wang and Jingjing Song carried out most of data analysis and drafted the manuscript. Yan Zhang designed and coordinated the study and revised the manuscript. Jinhua Song, Naiying Mao, Jiayuan Liang, Ye Chen, Ying Qi, Lina Bai, and Zhibo Xie performed the sequencing and sequence analysis. All authors read and approved the final manuscript.

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
