## [Reviewer comments · Microbiology Spectrum]

Microbiology Spectrum

An outbreak of severe neonatal pneumonia caused by human respiratory syncytial virus BA9 in a postpartum care centre in Shenyang, China

Bing Wang, Jingjing Song, Jinhua Song, Naiying Mao, Jiayuan Liang, Ye Chen, Ying Qi, Lina Bai, Zhibo Xie, and Yan Zhang

Corresponding Author(s): Yan Zhang, National Institute for Viral Disease Control and Prevention, China CDC

Review Timeline:

Submission Date:	March 20, 2022
Editorial Decision:	April 25, 2022
Revision Received:	May 9, 2022
Editorial Decision:	June 3, 2022
Revision Received:	June 7, 2022
Accepted:	June 18, 2022

Editor: Daniela Rajao

Reviewer(s): Disclosure of reviewer identity is with reference to reviewer comments included in decision letter(s). The following individuals involved in review of your submission have agreed to reveal their identity: Leyda Abrego (Reviewer #1); Hirokazu Kimura (Reviewer #2)

Transaction Report:

DOI: <https://doi.org/10.1128/spectrum.00974-22>

April 25, 2022

Dr. Yan Zhang
National Institute for Viral Disease Control and Prevention, China CDC
Beijing
China

Re: Spectrum00974-22 (An outbreak of severe neonatal pneumonia caused by human respiratory syncytial virus BA9 in a postpartum care centre in Shenyang, China)

Dear Dr. Yan Zhang:

Link Not Available

Sincerely,

Daniela Rajao

Journals Department
Reviewer comments:

Reviewer #1 (Comments for the Author):

You should focus on the genotype that you found in your samples and not conclude that the cause of outbreak are the nurses.

Reviewer #2 (Comments for the Author):

The authors studied molecular epidemiological analyses regarding an outbreak of pneumonia in neonate due to RSV-B, genotype BA9. Overall, the draft manuscript was well described, although subjects were relatively small numbers. I think that some major and

minor concerns should be improved.

1. The authors made a phylogenetic tree using NJ method. I think that this was made by ML method by MEGA.
2. Please provide the approval number of ethics committee.
3. Please provide analyzed nucleotide numbers of these sequences.
4. Please provide relevant discussion for amino acid substitutions in HRV2 in your strains comparing with other previous reports.

Staff Comments:

Preparing Revision Guidelines

Please return the manuscript within 60 days; if you cannot complete the modification within this time period, please contact me. If you do not wish to modify the manuscript and prefer to submit it to another journal, please notify me of your decision immediately so that the manuscript may be formally withdrawn from consideration by Microbiology Spectrum.

1 **An outbreak of severe neonatal pneumonia caused by human respiratory**
2 **syncytial virus BA9 in a postpartum care centre in Shenyang, China**

Bing Wang^{1,2*}, Jingjing Song^{1*}, Jinhua Song¹, Naiying Mao¹, Jiayuan Liang³, Ye
Chen², Ying Qi², Lina Bai², Zhibo Xie¹, Yan Zhang^{1#}.

¹National health commission (NHC) Key Laboratory of Medical Virology and
Viral Diseases, National Institute for Viral Disease Control and Prevention, China
CDC; WHO WPRO Regional Reference Measles/Rubella Laboratory.

²Shenyang Prefecture Center for Disease Control and Prevention, Shenyang, China

³Liaoning Prefecture Center for Disease Control and Prevention, Liaoning, China

* Bing Wang and Jingjing Song contributed equally to this paper.

**Keywords:** Human respiratory syncytial virus (HRSV); outbreak; neonatal
pneumonia; BA9 genotype; postpartum care centre

**Running title:** Outbreak of severe neonatal pneumonia caused by HRSV genotype
BA9

**Corresponding Authors:**

Yan Zhang, Email: zhangyan@ivdc.chinacdc.cn;

National health commission (NHC) Key Laboratory of Medical Virology and Viral

Diseases, National Institute for Viral Disease Control and Prevention, Chinese

Center for Disease Control and Prevention

No.155 Changbai road, Changping District, Beijing, China

**Abstract**

Human respiratory syncytial virus (HRSV) is a major pathogen of lower
respiratory tract infections in children (< 5 years) and older individuals, with
outbreaks mainly reported among infants in hospital paediatric departments and
intensive care units (ICUs). An outbreak of severe neonatal pneumonia occurred in
a postpartum centre in Shenyang city, China, from January to February, 2021. In
total, 34 respiratory samples were collected from 21 neonates and 13 nursing staff.
The samples were screened for 27 pathogens using a TaqMan low density array,
and 20 samples tested positive for HRSV, including 16 neonates and 4 nursing staff
samples. Among the 16 hospitalized neonates, seven were admitted to an ICU and
nine to general wards. Four of the nursing staff had asymptomatic infections. To
investigate the characteristics and source of the HRSV, the second hypervariable
region (HVR2) of the G gene of HRSV was sequenced from six neonates and two
nursing staff. Phylogenetic analyses revealed that eight of the sequences were
identical, clustering with the HRSV B subtype, close to the BA9 genotype
reference sequences, designated BA9-SY/CHN/2021. Subsequent genetic analysis
showed that BA9-SY/CHN/2021 belonged to lineage 7 of the BA9 genotype and
possessed five amino acid mutations compared with the BA9 reference strain
(GenBank: AB603467). In conclusion, this outbreak of severe neonatal pneumonia
was caused by HRSV genotype BA9, likely transmitted from the nursing staff. Our
findings highlight the critical need for strict hygiene and disease control measures

at such centres, to prevent outbreaks of severe infectious respiratory diseases.

**Importance**

Human respiratory syncytial virus (HRSV) is one of the leading causes of acute
lower respiratory infections (ALRI) worldwide, which is highly contagious and can
cause outbreaks in hospitals, military veteran centers, postpartum care centre
among infants, the elderly, and immunocompromised population. This is the first
reported an outbreak of neonatal severe clustered pneumonia caused by HRSVB
BA9 at a postpartum care centre in China, which leads to severe clinical symptoms
of ALRI such as cough, respiratory failure and even heart failure in neonates.
Based on the molecular epidemiological investigation, environmental monitoring
of the postpartum care centre, and working patterns of the nursing staff, it is
speculated that the outbreak may have been caused by transmission of HRSV
asymptomatic nurses to neonates.

**Keywords:** Human respiratory syncytial virus (HRSV); outbreak; neonatal
pneumonia; BA9 genotype; postpartum care centre

**Introduction**

Human respiratory syncytial virus (HRSV) is one of the most common
pathogens of acute lower respiratory infections (ALRI) in young children under 5

84 years of age, usually requiring hospitalization and mainly causing pneumonia and
85 bronchiolitis, and in even death severe cases (1, 2). The main manifestations of
86 ALRI caused by HRSV are a stuffy nose, runny nose and fever, cough, shortness of
87 breath, and respiratory failure(2). Its serious disease burden has aroused great
public concern. In 2015, an estimated 33.1 million children infected with HRSV-
related ALRI were reported worldwide, of which 3.2 million children were
hospitalized(3) .

HRSV is a negative-sense, single-stranded RNA virus of the genus
*Orthopneumovirus*, and the family *Pneumoviridae* (4). According to antigenic and
genetic differences, HRSV is divided into the HRSVA and HRSVB subtypes (5).
Based on the second hypervariable region (HVR2) located in the C-terminal
domain of the G gene, subtypes HRSVA and HRSVB can be further classified into
15 genotypes and 30 genotypes, respectively (6-9). In recent years, the BA9
genotype with a 60-bp repeat insertion in the G protein HVR2 region has gradually
become the predominant genotype globally(10).

Outbreaks of HRSV infection usually occur in general paediatric wards and
neonatal intensive care units (ICU) but have occasionally been reported in
postpartum care centre settings(1, 11-14). In recent years in China, an increasing
number of postpartum women seek professional care in postpartum care centre in
the first few months after discharge from maternity hospitals, so that neonates may
be exposed to some potential risk factors for respiratory disease infection in the

postpartum centres.

This study describes an outbreak of neonatal pneumonia in a postpartum care
centre in Shenyang city, Liaoning province, China, during January and February
2021. Clinical samples were collected from neonates and nursing staff to identify
the etiological agent and determine its origin. Screening of the samples indicated
that the outbreak was caused by HRSV. HVR2 fragments of the G gene of
Shenyang (SY) strains of HRSV were obtained to clarify the genetic characteristics
of the virus and the source of the virus infection.

**Materials and Methods**

**Ethics statement**

This study was approved by the second session of the Ethics Review
Committee of the National Institute for Viral Disease Control and Prevention of the
Center for Disease Control and Prevention (CDC) in China. Written informed
consent for the use of clinical specimens was obtained from all patients involved in
this study or their guardians. This study did not involve human experimentation;
the only human material used in this study was nasopharyngeal swab and throat
swab specimens collected from suspected ALRI cases during an outbreak in
Shenyang city of Liaoning province, China, from January to February, 2021.

**Specimen collection**

During the outbreak period, nasopharyngeal swab and throat swab were
collected from all neonates and nursing staff at the postpartum care centre. All
specimens were collected by epidemiology staff of Shenyang CDC and were
transported in sterile containers with a cold package (controlled low temperature of
4°C) to the Institute for Viral Disease Control and Prevention for further analysis.

**RT-PCR and sequencing**

The viral nucleic acid was directly extracted from the clinical specimens using
the TianLong nucleic acid extraction kit (Tianlong Biotechnology, Xian, China)
according to the manufacturer's instructions. The samples were screened for
human respiratory pathogens, including 16 viruses and 11 bacteria using multiplex
real-time RT-PCR with the TaqMan low density array (TLDA) kit (Thermo Fisher
Scientific Inc., Waltham, USA). The subtypes of HRSV were further identified by
in-house  real-time RT-PCR. The second hypervariable region (HVR2) (637–968
nt) of the G gene of HRSVB subtype was amplified using a one-step reverse
transcription-PCR kit (TaKaRa Biotechnology, Dalian, China) and the primer pair
GPB/F1 (15). The reaction conditions, as well as the purification and sequencing
protocols were as described previously(15, 16).

**Phylogenetic analysis**

Sequences were edited with Sequencher 5.0 (GeneCodes, Ann Arbor, MI,

USA). Multiple sequence alignments and pairwise distance were determined using
the MEGA program (Version 5.0; Sudhir Kumar, Arizona State University).
Phylogenetic trees were generated in MEGA using the neighbour-joining (NJ)
method and the maximum composite likelihood nucleotide substitution model.
Maximum likelihood (ML) phylogenetic trees were also generated. The reliability
of phylogenetic inference was estimated using the bootstrap method with 1000
replicates. Bootstrap values $\geq 70\%$ are shown.

**Nucleotide sequence accession numbers**

All sequences obtained in this study were submitted to the GenBank database
under the accession numbers OM892937-OM892944.

**Results**

**Epidemiological investigation**

From January 17 to February 3, 2021, an outbreak of neonatal pneumonia was
reported in a postpartum care centre in Shenyang city, China. In total, 16 out of 21
neonates from Shenyang postpartum care centre were hospitalized for clinical
treatment because of symptoms of ALRI. Among the 16 hospitalized ALRI cases,
seven neonates including premature twins were diagnosed with pneumonia and
admitted to the ICU, and nine neonates belonged to mild respiratory infection
cases. The average age of these 16 neonates was 23 days after birth (range: 12–28

168 days). In addition, 4 of 13 nursing staffs working in the postpartum care centre
during the outbreak belonged to lab-confirmed respiratory infection cases and the
average age of cases was 39 years (range: 24-47 years). No additional cases were
reported after February 3, 2021.

The index case of this outbreak (case No.1) occurred in the postpartum care
centre on January 17, 2021 with symptoms such as respiratory failure, and was
subsequently admitted to hospital after 3 days onset. Then, Case No. 2 in the same
postpartum care centre developed severe clinical symptoms 7 days after the onset
of case No.1, including a cough, shortness of breath, respiratory failure and
anemia. Cases No. 2 and 3 were twins that were born prematurely, and case No. 3
presented the similar symptoms with case No.2 except for heart failure.
Afterwards, case No. 4, 5, 6 and 7 from the same centre presented the same
symptoms as the above cases, while all the subsequent cases had relatively mild
symptoms, mainly manifested as coughing and nasal congestion (Table 1). Seven
of the neonates (case No.1 to No. 7) were transferred to the ICU because of heart
failure and respiratory failure, and all received intubation treatment. All neonatal
cases recovered after treatment in hospital.

**Etiological identification**

To identify the cause of this outbreak, pharynx swabs or nasopharyngeal
aspirates were collected from 21 neonates, while throat swabs were collected from

13 nursing staff. All of the samples were screened for 16 viruses, namely:
adenovirus; human bocavirus; parainfluenza virus; respiratory syncytial virus;
influenza virus; varicella zoster virus; Epstein–Barr virus; cytomegalovirus; human
herpesvirus 6; human metapneumovirus; measles virus; coronavirus 229E, HKU1,
NL63, OC43; mumps virus; enterovirus; rhinovirus; and human parecho virus; and
11 bacteria, namely: *Bordetella*; *Bordetella holmesii*; *Bordetella pertussis*;
*Chlamydomphila pneumoniae*; *Haemophilus influenzae*; *Klebsiella pneumoniae*;
*Legionella pneumophila*; *Moraxella catarrhalis*; *Mycoplasma pneumoniae*;
*Staphylococcus aureus* and *Streptococcus pneumoniae*. Twenty of the 34 samples
tested positive only for HRSV, and negative results were obtained for all other
viruses and bacteria. The 20 HRSV-positive samples were from 16 hospitalized
neonates and four nursing staff with asymptomatic infections. The other five
asymptomatic neonates and nine nursing staff were negative for all 27 pathogens.

**Characterization of HRSV associated with this outbreak**

The HVR2 fragment of G gene of HRSV was successfully amplified from six
neonates and two nursing staff. Genetic analysis revealed that these eight
sequences were 100% identical. Phylogenetic trees were constructed with the eight
sequences from this study and HRSV B reference sequences downloaded from the
GenBank database. These eight Shenyang (SY) sequences were clustered into the
same branch as the BA9 genotype references sequences. The sequences of the

viruses identified in this study were most closely related (98% homology) to
viruses detected in the Netherlands and Spain during 2018 and 2019 (Figure 1a).
The same results were obtained by performing a BLAST search of the SY
sequences against the GenBank database.

Next, 83 representative BA9 HVR2 sequences originating from 26 countries
during 2005 and 2019 were retrieved from the GenBank database. Phylogenetic
analysis was performed on the SY sequences and the 83 representative BA9 HVR2
sequences. The results showed that SY virus BA9-SY/CHN/2021 clustered with
lineage 7 of the BA9 genotype, which comprised viruses circulating in many
countries during 2017–2019 (Figure 1b, Table supplement).

Compared with the BA9 reference strain NG-102-06 (GenBank: AB603467),
there were five amino acid mutations (A271V, T276A, I281T, T290I and T312I) in
the HVR2 region of SY strain. The SY strain possessed a mutation in a stop codon
(from TAA to CAA) compared with the BA9 reference strain, resulting in a seven
amino acid extension to the G protein (Q-R-L-Q-S-Y-A).

**Discussion**

From January to February 2021, an outbreak of neonatal pneumonia caused
by a new lineage of the BA9 genotype of HRSV was reported in a postpartum care
centre in Shenyang, China. The outbreak resulted in the hospitalization of 16
neonates and of the nine severe cases, seven presented with respiratory failure,

including a pair of premature twins who also presented with heart failure. Based on
our epidemiological investigation, the transmission source for this outbreak may
have been the nursing staff who took care of the neonates.

The most common transmission routes of HRSV are via the respiratory tract
and direct contact, and the virus is able to survive on surfaces such as countertops
and cribs for hours (17). The HRSV incubation period ranges from 3 to 8 days
(18). Although the index case was identified among neonates, the nursing staff
with asymptomatic HRSV infections were considered as the possible source of this
outbreak, which could be supported by the following evidence. First, all of these
neonates were transferred from different maternity hospitals after birth and the
onset of disease in the index infant was 21 days after moving to the postpartum
care centre, which is much longer than the HRSV infection incubation period.
Therefore, it is likely that the index case was infected at the postpartum care centre
rather than the maternity hospital. Second, each neonate was cared for in a separate
room, but shared the same nursing staff. The nursing staff did not routinely wear
masks, and took care of different neonates without hand sanitation or changing
gowns. The viruses may therefore have been carried by the nursing staff and
transmitted from one neonate to another. Third, an investigation of the postpartum
care centre after the HRSV outbreak found that it is a relatively independent
building, covering an area of 800 m², with 28 rooms across three floors. Each room
contains a bathroom and a baby cot. There were no sterilization records for any

objects or rooms, the stored sanitizer had expired and the rooms were poorly
ventilated. In addition, the centre had no procedure in place for routine monitoring
of the health status of staff members. There was deemed a potential risk of indoor
cross-infection. Our nucleotide sequence analysis revealed that viral sequences
obtained from neonates and nursing staff were 100% identical. This indicated that
the HRSV associated with the pneumonia outbreak in neonates was originally
transmitted from nursing staff who had asymptomatic infections. It does not rule
out the transmission from the visiting family members, but unfortunately that no
samples have been collected and cannot be confirmed.

Genotype BA9 was first reported in 2006 and has since become the
predominant genotype worldwide(19) . According to a previous publication, the
sequences of BA9 genotype strains circulating worldwide from 2015–2019 could
be grouped into seven lineages, of which the 2017–2019 viruses belonged to
lineage 7 (20). Phylogenetic analysis in this study included the BA9 viruses
circulating worldwide from 2005–2019, and found that the virus associated with
the investigated outbreak clustered on the same branch as lineage 7 of the BA9
genotype. We detected a mutation in a stop codon of SY virus that resulted in a
seven amino acid extension to the G protein. This was consistent with a previous
report from The Observational United States Targeted Surveillance of Monoclonal
Antibody Resistance and Testing of HRSV (OUTSMART-RSV) study, which
analyzed viruses circulating in the USA during 2016 and 2017(21). The function of

these seven extended amino acids in the G protein remains to be determined, and
this extended G protein has not previously been linked to outbreaks of severe
cases. However, future research and surveillance to monitor the effect of this
genetic modification on viral epidemiology, transmission and disease severity is
warranted.

In conclusion, this outbreak of neonatal pneumonia was caused by the 2017–
2019 lineage of HRSV genotype BA9 and is believed to have been transmitted
from the nursing staff who cared for the neonates. Our findings highlight the
importance of strict hygiene and disease control measures, including
wearing/changing masks, hand washing, and changing gowns between neonates, to
prevent potential outbreaks of severe respiratory infectious diseases in such clinical
settings. To the best of our knowledge, this is the first report to describe a severe
neonatal pneumonia outbreak caused by HRSV of the BA9 genotype in a
postpartum care centre in China.

**Funding**

This work was supported by the National Health Commission Major Public
Health Project (ZDGW21-131031103000180005) .

**Conflict of interest**

The authors declare that no competing interests exist.

**Contributors**

BW and JJS carried out most of data analysis and drafted the manuscript. YZ
designed and coordinated the study and revised the manuscript. JHS, NYM, LJY,
CY, QY, BL and XZB performed the sequencing and sequence analysis. All authors
read and approved the final manuscript.

**References**

- 1. Moreno Parejo JC, Morillo García Á, Lozano Domínguez C, Carreño Ochoa C, Aznar Martín J, Conde Herrera
303 M. 2016. Respiratory syncytial virus outbreak in a tertiary hospital Neonatal Intensive Care Unit. *Anales de*
*Pediatría (English Edition)* 85:119-127.
- 2. Meissner HC, Ingelfinger JR. 2016. Viral Bronchiolitis in Children. *New England Journal of Medicine* 374:62-
72.
- 3. Shi T, McAllister DA, O'Brien KL, Simoes EAF, Madhi SA, Gessner BD, Polack FP, Balsells E, Acacio S, Aguayo
C, Alassani I, Ali A, Antonio M, Awasthi S, Awori JO, Azziz-Baumgartner E, Baggett HC, Baillie VL, Balmaseda
309 A, Barahona A, Basnet S, Bassat Q, Basualdo W, Bigogo G, Bont L, Breiman RF, Brooks WA, Broor S, Bruce
310 N, Bruden D, Buchy P, Campbell S, Carosone-Link P, Chadha M, Chipeta J, Chou M, Clara W, Cohen C, de
Cuellar E, Dang D-A, Dash-yandag B, Deloria-Knoll M, Dherani M, Eap T, Ebruke BE, Echavarria M, de Freitas
Lázaro Emediato CC, Fasce RA, Feikin DR, Feng L, et al. 2017. Global, regional, and national disease burden
estimates of acute lower respiratory infections due to respiratory syncytial virus in young children in 2015:
a systematic review and modelling study. *The Lancet* 390:946-958.
- 4. Rima B, Collins P, Easton A, Fouchier R, Kurath G, Lamb RA, Lee B, Maisner A, Rota P, Wang L, Ictv Report C.
2017. ICTV Virus Taxonomy Profile: Pneumoviridae. *J Gen Virol* 98:2912-2913.
- 5. Mufson MA, Orvell C, Rafnar B, Norrby E. 1985. Two distinct subtypes of human respiratory syncytial virus.
*J Gen Virol* 66 (Pt 10):2111-24.
- 6. Abrego LE, Delfraro A, Franco D, Castillo J, Castillo M, Moreno B, Lopez-Verges S, Pascale JM, Arbiza J.
2017. Genetic variability of human respiratory syncytial virus group B in Panama reveals a novel genotype
BA14. *J Med Virol* 89:1734-1742.
- 7. Cui G, Zhu R, Qian Y, Deng J, Zhao L, Sun Y, Wang F. 2013. Genetic variation in attachment glycoprotein
genes of human respiratory syncytial virus subgroups a and B in children in recent five consecutive years.
*PLoS One* 8:e75020.
- 8. Gaymard A, Bouscambert-Duchamp M, Pichon M, Frobert E, Vallee J, Lina B, Casalegno JS, Morfin F. 2018.
Genetic characterization of respiratory syncytial virus highlights a new BA genotype and emergence of the

ON1 genotype in Lyon, France, between 2010 and 2014. *J Clin Virol* 102:12-18.

9. Song J, Wang H, Ng TI, Cui A, Zhu S, Huang Y, Sun L, Yang Z, Yu D, Yu P, Zhang H, Zhang Y, Xu W. 2018.

Sequence Analysis of the Fusion Protein Gene of Human Respiratory Syncytial Virus Circulating in China

from 2003 to 2014. *Sci Rep* 8:17618.

10. Tabor DE, Fernandes F, Langedijk AC, Wilkins D, Lebbink RJ, Tovchigrechko A, Ruzin A, Kragten-Tabatabaie

332 L, Jin H, Esser MT, Bont LJ, Abram ME, Group I-RS. 2020. Global Molecular Epidemiology of Respiratory

Syncytial Virus from the 2017-2018 INFORM-RSV Study. *J Clin Microbiol* 59.

11. Halasa NB, Williams JV, Wilson GJ, Walsh WF, Schaffner W, Wright PF. 2005. Medical and economic impact

of a respiratory syncytial virus outbreak in a neonatal intensive care unit. *Pediatr Infect Dis J* 24:1040-4.

12. Mutlu M. 2015. Respiratory Syncytial Virus Outbreak Prevention by Screening Neonates with Respiratory

Infection, Isolation and Applying Standard Infection Control Procedures. *Neonatology & Clinical Pediatrics*

2:1-4.

13. Ryu S, Kim BI, Chun BC. 2018. An outbreak of respiratory tract infection due to Respiratory Syncytial Virus-

B in a postpartum center. *J Infect Chemother* 24:689-694.

14. Silva Cde A, Dias L, Baltieri SR, Rodrigues TT, Takagi NB, Richtmann R. 2012. Respiratory syncytial virus

outbreak in neonatal intensive care unit: Impact of infection control measures plus palivizumab use.

*Antimicrob Resist Infect Control* 1:16.

15. Zhang Y, Xu W, Shen K, Xie Z, Sun L, Lu Q, Liu C, Liang G, Beeler JA, Anderson LJ. 2007. Genetic variability of

group A and B human respiratory syncytial viruses isolated from 3 provinces in China. *Arch Virol* 152:1425-

34.

16. Song J, Wang H, Shi J, Cui A, Huang Y, Sun L, Xiang X, Ma C, Yu P, Yang Z, Li Q, Ng TI, Zhang Y, Zhang R, Xu W.

2017. Emergence of BA9 genotype of human respiratory syncytial virus subgroup B in China from 2006 to

2014. *Sci Rep* 7:16765.

17. Hall CB, Douglas RG, Jr., Geiman JM. 1980. Possible transmission by fomites of respiratory syncytial virus. *J*

*Infect Dis* 141:98-102.

18. Lessler J, Reich NG, Brookmeyer R, Perl TM, Nelson KE, Cummings DA. 2009. Incubation periods of acute

respiratory viral infections: a systematic review. *Lancet Infect Dis* 9:291-300.

19. Dapat IC, Shobugawa Y, Sano Y, Saito R, Sasaki A, Suzuki Y, Kumaki A, Zaraket H, Dapat C, Oguma T,

Yamaguchi M, Suzuki H. 2010. New genotypes within respiratory syncytial virus group B genotype BA in

Niigata, Japan. *J Clin Microbiol* 48:3423-7.

20. Chen X, Zhu Y, Wang W, Li C, An S, Lu G, Jin R, Xu B, Zhou Y, Chen A, Li L, Zhang M, Xie Z. 2021. A multi-

center study on Molecular Epidemiology of Human Respiratory Syncytial Virus from Children with Acute

Lower Respiratory Tract Infections in the Mainland of China between 2015 and 2019. *Virologica Sinica*

36:1475-1483.

21. Bin L, Liu H, Tabor DE, Tovchigrechko A, Qi Y, Ruzin A, Esser MT, Jin H. 2019. Emergence of new antigenic

epitopes in the glycoproteins of human respiratory syncytial virus collected from a US surveillance study,

2015-17. *Sci Rep* 9:3898.

Tables

Table 1. Characteristics of nine severe cases of human respiratory syncytial virus infection hospitalized in Shenyang Children's Hospital

Case No.	Age	Gender (M/F)	Data of disease onset	Pneumonia on CT scan	diagnose	Nasal	Cough*	Fever*	WBC	Respiratory failure	Heart failure	Hospitalized	ICU
27d	M	Jan 17	yes	ALRI	no	4d	no	$11.4 \times 10^9/L$	yes	no	yes	yes
26d	F	Jan 23	yes	ALRI	no	10d	no	$9.7 \times 10^9/L$	yes	no	yes	yes
26d	F	Jan 26	yes	ALRI	no	7d	no	$9.7 \times 10^9/L$	yes	yes	yes	yes
17d	M	Jan 28	yes	ALRI	4d	3d	no	$7.7 \times 10^9/L$	yes	no	yes	yes
28d	M	Jan 29	yes	ALRI	2d	2h	no	$9.9 \times 10^9/L$	yes	no	yes	yes
22d	M	Jan 30	yes	ALRI	3d	3d	no	$9.2 \times 10^9/L$	yes	no	yes	yes
12d	M	Feb 1	yes	ALRI	no	2d	no	$9.2 \times 10^9/L$	yes	no	yes	yes
27d	M	Jan 30	yes	ALRI	1d	4d	1d	$12.93 \times 10^9/L$	no	no	yes	no
23d	M	Feb 3	yes	ALRI	2d	2d	no	$6.3 \times 10^9/L$	no	no	yes	no

*The number of days

M, male; F, female; HRSV, human respiratory syncytial virus; WBC, white blood cells.

Figure

Figure 1. Neighbor-joining phylogenetic tree of the entire coding region nucleotide sequence of HVR2 of the G gene of HRSV subgroup B isolated from the postpartum care centre. Red circles represent sequences from neonates; red triangles represent sequences from nursing staff. (b) Phylogenetic analysis of the sequences of BA9-SY/CHN/2021 isolates from the outbreak in Shenyang (shown in red) and 83 genotypes of BA9 strains (2005–2019) retrieved from the GenBank database. Blue triangles indicate representative strains from China.

Following are the point-by-point responses to reviewers

Reviewer #1 (Comments for the Author):

You should focus on the genotype that you found in your samples and not conclude that the cause of outbreak are the nurses.

Response:

Thank you very much for your valuable comments, we really appreciated the time that you spent in reviewing our manuscript. We agreed with you and have already revised the manuscript as you suggested.

Minor comments:

Line 57, I think that a new name should not be given to the BA9 strain found.

Response:

Thank you very much for your comments. Sorry for the confusing name. We used the name of “SY strain” to refer the viruses identified in this study (line 58).

Line 97, You should eliminate bibliography 9 because it talks about the second hypervariable region of the G gene and citation 9 corresponds to a study of the F gene.

Response:

Thank you very much for your suggestion. As suggested, we deleted the bibliography 9.

Line 111-113, Because it refers to a particular strain, Shenyang (SY)?

Response:

Sorry for the confusing name. Shenyang (SY) strains refers to “SY strain”. To avoid the confusion, we used the name of “SY strain” to refer the viruses identified in this study, in the whole manuscript.

Line 141, You must describe the methodology because it has not been previously published or is not cited.

Response:

Thank you very much for your comments. As suggested, we referred the reference of the real-time RT-PCR methodology in the revised manuscript (line 141).

Line 264-267, Due to the fact that it was not possible to obtain a sample from the relatives, it cannot be concluded that the outbreak was caused by the nurses.

Response:

Thank you very much for your comments. We agree with you. So we revised the conclusion in the manuscript: It is inferred that this outbreak of neonatal pneumonia **might be** spread by nursing staffs with asymptomatic infections, but could not rule out the visiting relatives who might bring the HRSV virus to the postpartum care centre. Unfortunately, the samples were not available from the visiting relatives so that no evidence to support this speculation. Please see the modification in the line of 264-267.

Reviewer #2 (Comments for the Author):

The authors studied molecular epidemiological analyses regarding an outbreak of pneumonia in neonate due to RSV-B, genotype BA9.

Overall, the draft manuscript was well described, although subjects were relatively small numbers. I think that some major and minor concerns should be improved.

Response:

We really appreciated your positive comments on our manuscript, thank you very much for your expertise and your valuable comments.

1. The authors made a phylogenetic tree using NJ method. I think that this was made by ML method by MEGA.

Response:

After double check, we found the phylogenetic tree analysis showed in this manuscript was made by NJ method of MEGA, rather than ML method.

2. Please provide the approval number of ethics committee.

Response:

Thank you very much for your comments. As suggested, we provided the approval number of the ethics committee is IVDC 2018 No. 012 (line 119-120).

3. Please provide analyzed nucleotide numbers of these sequences.

Response:

Thank you very much for your comments. As suggested, we provided analyzed nucleotide numbers (324 nucleotide) of eight sequences in the manuscript.

Please see the line 206 in the revised manuscript as below: The HVR2 fragment of G gene of HRSV (324 nucleotide) was successfully obtained from six neonates and two nursing staff respectively.

4. Please provide relevant discussion for amino acid substitutions in HRV2 in your strains comparing with other previous reports.

Response:

Thank you very much for your comments. Compared with the previous reports, we supplemented the relevant contents of amino acid mutation in HRV2 in the discussion as below: amino acid analysis of the SY strains identified in this study showed there were five mutations in the HVR2 of the G protein gene, including A271V, T276A, I281T, T290I and T312I. Previous studies have found the same amino acid mutation sites, and five simultaneous amino acid mutation sites were found in six identical sequences from patients hospitalized with community acquired pneumonia (CAP) patients from China during 2018. However, whether these five amino acid mutations

found in the viruses associated with severe neonatal pneumonia could be beneficial for the virus replication and transmission remains to be further studied (line 274-282).

June 3, 2022

Dr. Yan Zhang
National Institute for Viral Disease Control and Prevention, China CDC
Beijing
China

Re: Spectrum00974-22R1 (An outbreak of severe neonatal pneumonia caused by human respiratory syncytial virus BA9 in a postpartum care centre in Shenyang, China)

Dear Dr. Yan Zhang:

Link Not Available

Sincerely,

Daniela Rajao

Journals Department
Reviewer comments:

Reviewer #1 (Comments for the Author):

After reviewing the responses to my comments, I found that they were all taken into consideration and included in the text.

Reviewer #2 (Comments for the Author):

NJ is not well regarded among evolutionary biologists as its methods are too ad hoc. Moreover, this method is not suitable for rapid evolutionary gene including the G gene. The author should remake the phylogenetic tree using ML method.

Second, the authors used lineage in the phylogenetic tree. In general, the term lineage or clade usually can use large cluster having large genetic divergence (i.e., corresponded phylogenetic distance at more than around 0.2 or 0.3. The present strains probably had less than 0.1. Thus, the suitable term is cluster.

Staff Comments:

Preparing Revision Guidelines

Please return the manuscript within 60 days; if you cannot complete the modification within this time period, please contact me. If you do not wish to modify the manuscript and prefer to submit it to another journal, please notify me of your decision immediately so that the manuscript may be formally withdrawn from consideration by Microbiology Spectrum.

Following are the point-by-point responses to reviewers

Reviewer #1 (Comments for the Author):

After reviewing the responses to my comments, I found that they were all taken into consideration and included in the text.

Thank you for your review again.

Reviewer #2 (Comments for the Author):

NJ is not well regarded among evolutionary biologists as its methods are too ad hoc. Moreover, this method is not suitable for rapid evolutionary gene including the G gene. The author should remake the phylogenetic tree using ML method.

Second, the authors used lineage in the phylogenetic tree. In general, the term lineage or clade usually can use large cluster having large genetic divergence (i.e., corresponded phylogenetic distance at more than around 0.2 or 0.3. The present strains probably had less than 0.1. Thus, the suitable term is cluster.

Response:

Thank you very much for your expertise and your valuable comments. The phylogenetic tree has been reconstructed using ML method. In addition, we changed “lineage” to “cluster” and modified it in the whole manuscript.

June 18, 2022

Dr. Yan Zhang
National Institute for Viral Disease Control and Prevention, China CDC
Beijing
China

Re: Spectrum00974-22R2 (An outbreak of severe neonatal pneumonia caused by human respiratory syncytial virus BA9 in a postpartum care centre in Shenyang, China)

Dear Dr. Yan Zhang:

Your manuscript has been accepted, and I am forwarding it to the ASM Journals Department for publication. You will be notified when your proofs are ready to be viewed.

Sincerely,

Daniela Rajao
Editor, Microbiology Spectrum

Journals Department
Supplemental file 1: Accept